# Leveraging heterogeneity across multiple datasets increases cell-mixture deconvolution accuracy and reduces biological and technical biases

Francesco Vallania [1,2], Andrew Tam[1,3], Shane Lofgren[1,2], Steven Schaffert[1,2], Tej D. Azad[1], Erika Bongen[1], Winston Haynes [2], Meia Alsup[1,3], Michael Alonso[4], Mark Davis[1], Edgar Engleman[4] & Purvesh Khatri [1,2]

In silico quantification of cell proportions from mixed-cell transcriptomics data (deconvolution) requires a reference expression matrix, called basis matrix. We hypothesize that matrices created using only healthy samples from a single microarray platform would introduce biological and technical biases in deconvolution. We show presence of such biases in two existing matrices, IRIS and LM22, irrespective of deconvolution method. Here, we present immunoStates, a basis matrix built using 6160 samples with different disease states across 42 microarray platforms. We find that immunoStates significantly reduces biological and technical biases. Importantly, we find that different methods have virtually no or minimal effect once the basis matrix is chosen. We further show that cellular proportion estimates using immunoStates are consistently more correlated with measured proportions than IRIS and LM22, across all methods. Our results demonstrate the need and importance of incorporating biological and technical heterogeneity in a basis matrix for achieving consistently high accuracy.

[1] Institute for Immunity, Transplantation and Infection, Stanford University, Stanford 94305 CA, USA. [2] Stanford Center for Biomedical Informatics Research, Department of Medicine, Stanford University, Stanford 94305 CA, USA. [3] Stanford Institutes of Medicine Summer Research Program, Stanford University, Stanford 94305 CA, USA. [4] Department of Pathology, Stanford University, Stanford 94305 CA, USA. Correspondence and requests for materials should be addressed to P.K. (email: pkhatri@stanford.edu)

Cell-mixture deconvolution is an established in silico approach for quantifying cell subpopulations directly from bulk gene expression data of mixed cell samples[1–4]. Multiple computational methods have been developed[5,6] to estimate the proportions of immune cells in blood[6] and tissue biopsies[7], as well as cell-type specific expression profiles from bulk expression data[8,9]. The underlying assumption in virtually every deconvolution approach to date is that the observed expression of any given gene in a mixed-tissue sample is a combination of its expression across each cellular subset[1]. Based on this assumption, methods for estimating cellular frequencies from mixed tissue data use a variant of a regression model, such as linear regression[3], quadratic programming[4], robust regression[7], or support vector regression[7]. Irrespective of the type of statistical model, each method requires a reference expression matrix, called a basis matrix, that is composed of genes specifically expressed in the expected cell subsets found in the tissue of interest[8].

Typically, a basis matrix is constructed from sorted cell expression data by combining expression profiles of sorted immune cells from one (e.g., IRIS[2,3]) or more datasets (e.g., LM22[7]), which are profiled using a single microarray platform to ensure homogeneity in expression data. However, it is possible that this approach can introduce a technical bias in a basis matrix towards the microarray platform used for transcriptome profiling, resulting in lower deconvolution accuracy for samples that are profiled using different platforms. Furthermore, basis matrices to date have been created using expression data solely from healthy subjects[2,3,7], which can further introduce biological bias that could affect deconvolution accuracy and limit their applicability to samples from patients with disease.

Here, we show that current deconvolution approaches are significantly affected by technical and biological bias by measuring accuracy across 5540 human transcriptomes. We find that the presence of these biases substantially reduces deconvolution accuracy. We therefore hypothesized that a basis matrix created by integrating data from multiple independent cohorts of healthy and disease samples profiled using different microarray platforms would reduce biological and technical biases and improve accuracy of deconvolution. To test this hypothesis, we present a new basis matrix, immunoStates, that leverages biological and technical heterogeneity across 6160 whole transcriptomes of human sorted blood cells measured on 42 microarray platforms. We use a multi-cohort analysis framework that leverages biological and technical heterogeneity present across multiple independent studies for this purpose. This approach has been previously shown to increase reproducibility in gene expression signatures across a broad spectrum of diseases including organ transplant, sepsis, infectious diseases, cancer, and systemic sclerosis[10–20]. We show that immunoStates allows for more accurate deconvolution with substantially reduced bias across different microarray platforms and disease samples. Importantly, our analyses show that, for any given basis matrix, different deconvolution methods produce highly correlated results, demonstrating that the choice of the matrix is more important than the deconvolution method itself. We also find that the accuracy of a basis matrix is strongly dependent on its set of signature genes rather than the expression values of the gene themselves. Our findings provide strong evidence for the importance of the basis matrix in determining deconvolution accuracy. We conclude that incorporation of technical and biological heterogeneity in the construction of the matrix reduces bias, which increases accuracy in cell mixture deconvolution independently of the method.

## Results

**Single microarray platform basis matrices contain technical bias.** We hypothesized that a basis matrix created using gene expression data generated from a single microarray platform would exhibit significant platform-dependent bias (technical bias) in deconvolution accuracy. To test our hypothesis, we used IRIS and LM22 as basis matrices, both of which are constructed using only healthy samples profiled only on Affymetrix microarrays[3,7]. We deconvolved 17 independent datasets consisting of 1071 whole transcriptome profiles of human peripheral blood mononuclear cells (PBMCs) measured across eight microarray platforms from two different manufacturers (see Methods and Supplementary Table 1) using both basis matrices. We defined this set as a "technical bias evaluation cohort". Further, to generalize our findings across multiple methods, we used five deconvolution algorithms (linear regression, PERT, quadratic programming, robust regression, and support vector regression)[3,4,7,21]. We estimated accuracy of deconvolution across samples by computing their goodness of fit as previously described (see Methods section)[7]. Briefly, if the original mixed-tissue sample expression could be reconstituted by combining the estimated proportions of individual cell types with the expression from basis matrix, we would observe high goodness of fit and expect good deconvolution accuracy.

We observed significant differences in goodness of fit between microarray platforms for both matrices, irrespective of the method used (Fig. 1a, b), demonstrating the presence of platform-specific bias. We quantified the extent of these differences for each basis matrix using median absolute deviation (MAD) of goodness of fit, a measure of heterogeneity robust to outliers as described before[22], across samples from different platforms. We calculated MAD as difference in goodness of fit for each sample from mean goodness of fit across all platforms for a given basis matrix, and estimated its statistical significance against the null hypothesis that there was no technical variation between samples (see Methods section). We observed significant heterogeneity in goodness of fit between platforms for both IRIS (MAD = 0.21, $p = 2.71e{-}8$) and LM22 (MAD = 0.09, $p = 4.4e{-}2$), irrespective of the method used (Supplementary Fig. 1).

Arguably, support vector regression re-scales expression data prior to deconvolution, which is not required for other methods such as linear model and robust regression. Rescaling gene expression data when using these methods could potentially reduce their accuracy. Therefore, we compared estimated proportions for linear model and robust regression with and without rescaling gene expression data in the technical bias evaluation cohort using IRIS and LM22 as basis matrices. We found there was very high correlation in estimated cellular proportions suggesting rescaling did not adversely affect linear model and robust regression (Supplementary Fig. 2). Based on these results, we chose to maintain a uniform preprocessing strategy across all methods for the rest of the manuscript.

**Leveraging heterogeneity reduces technical bias in a basis matrix.** Next, we hypothesized that a basis matrix created using multiple microarray platforms will reduce platform-dependent technical bias in cellular deconvolution. We collected 165 publicly available gene expression datasets from GEO that profiled 6160 samples from 20 sorted human blood cell types using 42 microarray platforms (Supplementary Fig. 3A and Supplementary Data 1, see Methods section). We did not discard experiments based on sorting strategy, platform manufacturer, or disease state of the sample.

Using these data, we created a new basis matrix consisting of 317 cell-type specific genes, called immunoStates, (Supplementary

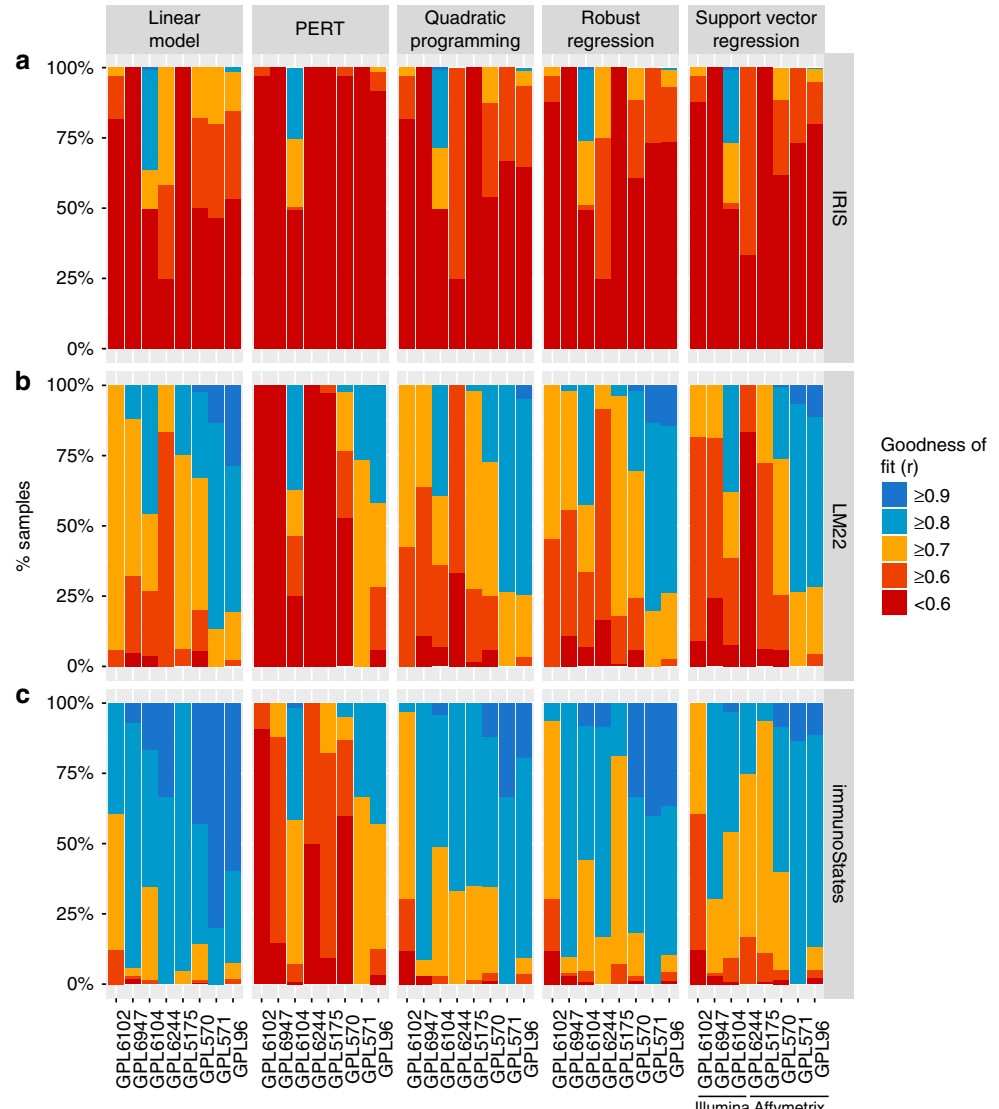

**Fig. 1** Analysis of platform bias in deconvolution across multiple methods and matrices. **a** Goodness of fit values across 1071 human PBMC samples as a function of microarray platform using the IRIS signature matrix. Goodness of fit is displayed as a stacked barplot with color indicating corresponding values starting from goodness of fit value of 0.5 or lower up to values of 0.9 and above. Barplots are grouped by the method of deconvolution used for the analysis. **b** Same as in **a** for LM22. **c** Same as in **a** for immunoStates

Fig. 3B and Supplementary Data 2, see Methods section). A large fraction of genes in immunoStates (76%) was not shared with IRIS or LM22 (Supplementary Fig. 3C). We then deconvolved the technical bias evaluation cohort using immunoStates as a basis matrix across all five methods. Unlike IRIS and LM22, there was no heterogeneity in goodness of fit between microarray platforms for immunoStates (Fig. 1c; MAD = 0.07, $p = 0.16$; Supplementary Fig. 1). Importantly, mean goodness of fit using immunoStates was significantly higher than IRIS and LM22 ($p < 2.2e-16$) irrespective of the method used.

Arguably, the higher goodness of fit of immunoStates could be due to the higher amount of data used to create it. It is possible that if the same amount of data were used to create one of the existing basis matrices, it would have higher goodness of fit as well. We investigated this argument by modifying LM22 such that it contained the same genes, but their expression values were computed using the data sets used for creating immunoStates. We found that despite increasing the amount of data used to estimate expression values for LM22 genes, it continued to have platform bias as before with significant heterogeneity in goodness of fit

(MAD = 0.05, $p = 0.05$, Supplementary Fig. 4). These results suggest that better estimation of expression values for genes in a basis matrix using large amount of data is not sufficient to increase deconvolution accuracy. Further, these results strongly suggest that selection of genes in a basis matrix from biologically and technologically heterogeneous data is more important in reducing bias. Together, our results demonstrate that a basis matrix created using heterogeneous data from multiple platforms reduces technical bias.

**Including disease samples in a basis matrix reduces biological bias.** Cell quantification technologies, such as FACS and CyTOF, use a set of predefined phenotypic markers that are not affected by either disease- or treatment-induced changes[23]. For instance, irrespective of whether a sample is from a healthy control or a patient with or without any treatment, *CD14* and *CD56* are used to identify monocytes and natural killer cells, respectively. Similarly, a basis matrix should be unaffected by disease- and treatment-induced changes to be broadly applicable across a large number of diseases and conditions.

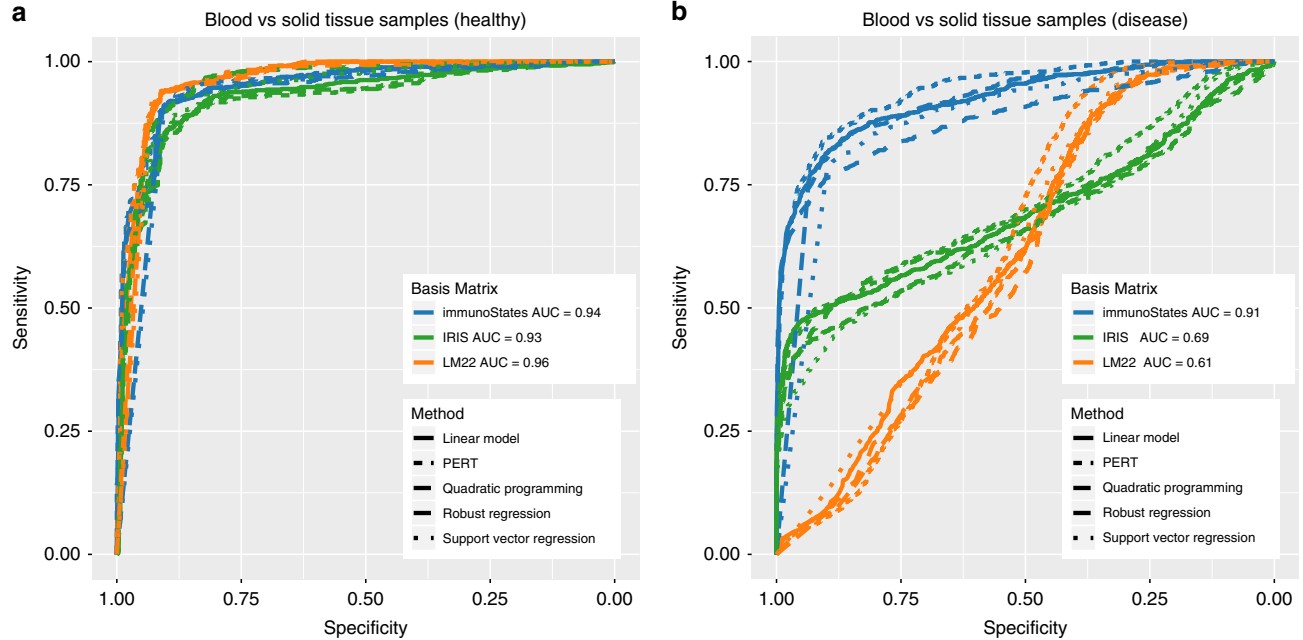

**Fig. 2** Effect of disease on deconvolution. **a** ROC curves indicating the ability of IRIS, LM22, and immunoStates (denoted by line color) to distinguish blood-derived samples from tissue biopsies in healthy donors (1383 samples) using goodness of fit across all tested methods (denoted by line type). AUCs indicate mean AUC for an individual signature matrix across all methods. **b** Same as in **a** but in disease samples (2684 samples)

We hypothesized that a basis matrix created using only healthy samples (e.g., IRIS and LM22) will have lower goodness of fit when deconvolving a disease sample, and hence, lower deconvolution accuracy, whereas a basis matrix created using both healthy and disease samples (e.g., immunoStates) will have higher goodness of fit and accuracy. To test this hypothesis, we used E-MTAB-62 [https://www.ebi.ac.uk/arrayexpress/experiments/E-MTAB-62/], a gene expression compendium of 5372 samples representing primary tissues and cell lines[24]. For the purpose of this analysis, we considered only primary samples from human subjects, consisting of 4067 blood-derived and tissue-derived samples from either healthy donors or individuals with a disease, such as leukemia, solid tumors, and neurodegenerative disorders. For each pair-wise combination of a basis matrix and a deconvolution method, we determined the effect of disease on deconvolution accuracy by estimating its ability to distinguish blood- from tissue-derived samples based on significance of goodness of fit, as described previously[7].

Because each of the three matrices only contained blood cells, we would expect them to have significantly higher goodness of fit values for blood-derived samples compared to those for solid tissue samples as they are not represented in any basis matrix. In 1383 healthy samples in E-MTAB-62, irrespective of the method, the goodness of fit was higher for blood-derived samples than tissue-derived samples for all basis matrices (Supplementary Fig. 5A). This result translated into an accurate distinction of blood from tissue-derived samples for all matrices and methods based on significance of deconvolution (AUCs: IRIS 0.9335 ± 0.0001; LM22 0.9589 ± 0.0001; immunoStates 0.9414 ± 0.0001, Fig. 2a) for healthy samples.

Further, if the expression of genes in a basis matrix changed in a disease sample, we would expect low goodness of fit for both blood and solid tissue samples, indicating lower deconvolution accuracy. For 2684 disease samples in E-MTAB-62, when using IRIS or LM22, the goodness of fit values for blood-derived and tissue-derived samples were highly similar (Supplementary Fig. 5B) and resulted in lower discrimination between them (IRIS: AUC = 0.6908 ± 0.0001, LM22: AUC = 0.6123 ± 0.0001;

Fig. 2b). In contrast, immunoStates had significantly higher goodness of fit for blood-derived samples than tissue-derived samples, irrespective of the deconvolution method used, resulting in high accuracy for distinguishing blood- and tissue-derived disease samples (AUC = 0.9081 ± 0.0001; Fig. 2b and Supplementary Fig. 5B). Collectively, these results demonstrated that a basis matrix created using only healthy transcriptome profiles contains a biological bias against disease samples, which makes it difficult to distinguish between blood and solid tissue samples, and results in lower deconvolution accuracy. In contrast, creating a basis matrix using both healthy and disease samples significantly reduces the biological bias and increases deconvolution accuracy.

**Different methods produce highly correlated results for a given matrix.** Our results revealed that incorporating biological and technical heterogeneity by using both healthy and disease samples profiled across multiple platforms in a basis matrix reduced platform and disease bias irrespective of the deconvolution method used. Therefore, we tested whether the basis matrix had a stronger effect on the deconvolution results than the method used to estimate cell proportions. For a given basis matrix, all methods produced highly correlated cell proportion estimates ($r = 0.762 ± 0.014$ Fig. 3). In contrast, for a given method, we observed significantly lower correlations in cell proportion estimates when using different basis matrices ($r = 0.452 ± 0.029$, $p = 2.7e−15$), or when both matrix and method were different ($r = 0.451 ± 0.015$, $p < 2.2e−16$). We observed these trends irrespective of whether the sample came from blood or solid tissue biopsies (Supplementary Fig. 6). These results provide a strong evidence that the basis matrix is the major determinant of deconvolution accuracy, and demonstrate that virtually no method can overcome biological and technical bias present in a basis matrix.

**Reducing bias in a basis matrix increases accuracy of deconvolution.** Despite demonstrated utility of goodness of fit in evaluating accuracy of deconvolution[7], cell count data from

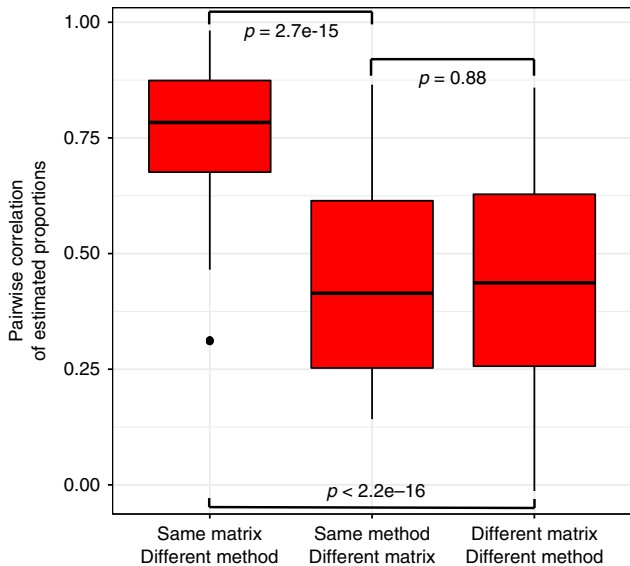

**Fig. 3** Deconvolution concordance by matrix and method. Boxplots represent the distribution of pairwise correlation coefficients between estimated proportions for all matrices and deconvolution methods. Center lines correspond to the median value of each box and the lower and upper bounds of each box correspond to their first and the third quartiles, respectively. Comparisons were divided in (1) pairs with the same signature matrix but run with different methods, (2) pairs with different signature matrices but run using the same method, and (3) pairs where both matrix and method were different. Significance analysis was performed using the Wilcoxon's paired rank sum test

high-resolution technologies such as FACS is the most appropriate way to evaluate the performance of deconvolution algorithms. Therefore, we explored whether the reduction in biological and technical bias in a basis matrix results in increased accuracy of deconvolution by correlating estimated cell proportions with cell proportions measured using FACS or Coulter counter for each pair of a basis matrix and a deconvolution method. We identified five gene expression datasets of 402 human whole blood or PBMC samples with paired cell counts data available (see Methods section). These datasets were generated using Illumina HT12 V4.0 or Affymetrix Primeview microarrays as follows: (1) two independent datasets consisting of 176 healthy human PBMC samples profiled using Illumina HT-12 V4.0 arrays paired with flow-cytometry data (GSE65133, GSE59654)[7,25], and (2) a whole blood dataset of 226 healthy samples, a subset of which were profiled over three consecutive years using Affymetrix PrimeView arrays (see Methods section).

Across the five datasets, estimated cell proportions by IRIS and LM22 had significantly lower correlations with measured cell proportions (IRIS: $r = 0.10 \pm 0.06$, $p = 1.1e{-}6$; LM22: $r = 0.57 \pm 0.05$, $p = 6.1e{-}3$) compared to immunoStates ($r = 0.74 \pm 0.04$) (Fig. 4a, Supplementary Figs 7–14). In concordance with a previous report, we found that IRIS and LM22 systematically over-estimated or under-estimated individual cellular proportions even for high frequency cell subsets such as Monocytes and CD4 + T-cells (Supplementary Fig. 15)[7]. Importantly, across all basis matrices, no method produced consistently higher correlations with measured cell proportions (p≥0.25), and provided further evidence that the accuracy of deconvolution is determined by a basis matrix instead of a deconvolution method.

We then quantified the extent of over-estimation and under-estimation of cell proportions across all cohorts and methods by computing the root mean square error (RMSE) between measured and estimated proportions (Fig. 4b). We found that

immunoStates generated estimates with significantly lower RMSE ($RMSE = 12.77 \pm 0.73$) than IRIS ($RMSE = 28.17 \pm 2.16$, $p = 2.6e{-}5$) and LM22 ($RMSE = 16.79 \pm 1.12$, $p = 4.1e{-}3$) across all cohorts and deconvolution methods used. Again, no single method had significantly lower RMSE than others across all matrices and cohorts (p≥0.40). Our results provide strong evidence that accuracy of current deconvolution methods is affected by biological and technical biases present in a basis matrix, and no method is able to overcome these biases.

Overall, our results suggest that a one-size-fits-most basis matrix may be a more robust solution for estimating cellular proportions than creating a platform-specific basis matrix. Therefore, we investigated whether a basis matrix created using multiple datasets from a single microarray platform would be as accurate as immunoStates. We generated two new basis matrices: (1) using data only from a specific Illumina microarray (GPL10558) and (2) using data from all Illumina microarrays. We deconvolved GSE65133 [https://www.ncbi.nlm.nih.gov/geo/query/acc.cgi?acc=GSE65133], which was profiled using GPL10558, using these two Illumina-specific basis matrices and immunoStates, and correlated estimated cellular proportions with those measured by cytometry. Compared to immunoStates, both Illumina-specific basis matrices had lower correlations, irrespective of the method used (Supplementary Fig. 16).

Collectively, our results demonstrate that creating a basis matrix by leveraging biological and technical heterogeneity across multiple independent cohorts reduces bias, and significantly improves accuracy of deconvolution irrespective of the statistical model used.

## Discussion

Using whole transcriptome profiles, cell-mixture deconvolution methods quantify cell subsets within a mixed-tissue sample without physical separation of its components. We hypothesized that the current practice of creating a basis matrix for deconvolution using only healthy samples profiled using the same microarray platform introduces biological and technical biases that reduce accuracy of deconvolution. Our analysis of two basis matrices, IRIS and LM22, both created using only one microarray platform and healthy samples, showed significant heterogeneity in deconvolution results between different microarray platforms, and lower discriminatory power for distinguishing blood and solid tissue samples when obtained from a patient instead of a healthy control.

There is increased evidence that leveraging biological and technical heterogeneity across multiple independent datasets identifies robust and reproducible gene signatures[10–20]. Here, we hypothesized that the heterogeneity present in publicly available datasets can be used to create a basis matrix with significantly reduced biological and technical bias, and increase accuracy of deconvolution. We used 165 publicly-available gene expression datasets that profiled 6160 sorted human immune cell samples using 42 different microarray platforms to create a 317-gene basis matrix called immunoStates. Our analysis showed that immunoStates substantially reduced technical and biological bias and resulted in more accurate cell proportion estimates. Unexpectedly, we found that the accuracy of all basis matrices was independent of the statistical method used for deconvolution. For a given basis matrix, all methods produced highly correlated cellular proportions. We also found that using more data to estimate expression values for the same set of genes in a basis matrix continued to maintain the platform bias, and did not improve deconvolution accuracy.

These results have important long-term implications as virtually all efforts to date have been focused on developing new methods to improve deconvolution accuracy rather than the

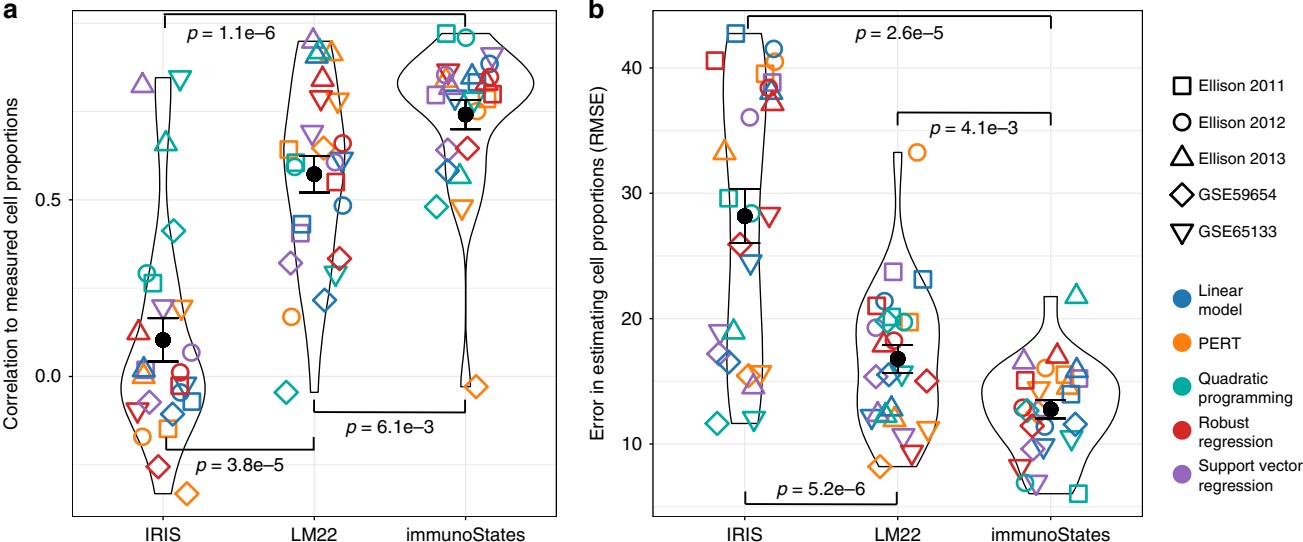

**Fig. 4** Correlation with measured cell proportions across 402 human blood samples. **a** Correlation between measured cell proportions and deconvolution estimates in five different human sample cohorts (denoted by different shapes) across different deconvolution methods (denoted by different colors) using IRIS, LM22, and immunoStates (*x*-axis). Correlation is measured by Pearson's correlation coefficient. Center dot represents mean value for each violin plot. Error bars represent standard error of the mean. **b** Same as in **a** for RMSE between measured and estimated cell proportions. Significance analysis was performed using the Wilcoxon's paired rank sum test

design of the basis matrix[6]. While the choice of a method may be important in specific use cases, our results argue improvements on the basis matrix are beneficial across all methods. Improvements in methods are usually demonstrated in particular contexts, such as estimating proportions in the presence of background tissue[7] or robustness to expression changes due to treatments[21]. The design and execution of these studies typically involve the application of the same basis matrix, and result in improvements in accuracy provided by the new method for the dataset that represent the use case of interest. While the choice of method is therefore important in such specific cases, our results strongly support the argument that improvements on the basis matrix improve accuracy across all methods and datasets.

Our results across multiple independent platforms also suggest that immunoStates may be a one-size-fits-most basis matrix readily applicable to future deconvolution methods that rely on the use of a basis matrix to estimate cell proportions. This strategy has a number of practical advantages. First, researchers could directly apply immunoStates without having to select a basis matrix most appropriate for their dataset of interest. Second, the application of a single basis matrix facilitates comparison and integration of cell proportions from multiple datasets across multiple platforms and different centers. Third, we believe that instead of replacing cytometry-based experimental approaches such as FACS or CyTOF, statistical deconvolution of mixed-tissue expression profiles will complement these technologies. Current cytometry approaches are limited in the number of variables they can profile simultaneously due to the number of channels available for measurements. a one-size-fits-most approach such as immunoStates is simpler from a practical standpoint as a researcher could directly apply without having to select the matrix most appropriate for their dataset of interest. The researchers can then design a panel of cell type markers for a mass cytometry experiment that is focused on the cell types identified through statistical deconvolution of transcriptome data. Such a strategy would reduce wasting of channels by allowing the researchers to eliminate markers unlikely to show any differences, and better utilize the available channels by including markers that allow better phenotyping.

Our approach has a few limitations. First, despite more than 1 million human transcriptome profiles in NCBI GEO and EBI ArrayExpress, more specific and rare cell subsets are less likely to have sufficient heterogeneous data across multiple platforms and biological conditions. It is not clear how much data is sufficient to represent cellular heterogeneity. Our previous work has suggested that 4–5 datasets consisting of approximately 250 samples may be enough to represent disease heterogeneity[17]. However, the heterogeneity at cellular level may require more datasets. We expect that continued accumulation of additional sorted-cell datasets in public repositories over time will increase both the accuracy and the breadth of future basis matrices. Furthermore, increased availability of single cell RNAseq data will further facilitate creation of better and more refined basis matrices.

Second, we emphasize that immunoStates did not remove these biases entirely, but reduced them substantially, which in turn significantly improved accuracy of deconvolution. We expect that these biases will continue to reduce as more data for sorted cells becomes available and are used to update immunoStates in the future.

Third, all basis matrices require a priori knowledge of the populations within the sample of interest. However, current cytometry-based methods also require a priori selection of markers. We expect these limitations will be overcome as more single-cell transcriptomic data become available. These data will allow for the discovery of previously unknown cell subsets that can be included in a basis matrix. These data will also increase heterogeneity and the sample size, which based on our analysis presented here, will further improve accuracy of deconvolution. The advantage of immunoStates is that through increased accuracy and reduced technical and biological bias, independent of the method used, it can leverage existing data in public repositories to identify cellular subsets that should be further explored using targeted technologies such as FACS or CyTOF. By avoiding cellular subsets that are not changing in existing data and avoiding further profiling them, immunoStates can help researchers design better experiments that increase the probability of identifying relevant and novel cell subsets.

## Methods

**Dataset collection and pre-processing.** Unless otherwise noted, we downloaded all datasets from Gene Expression Omnibus (GEO, www.ncbi.nlm.nih.gov/geo/) using the MetaIntegrator package from CRAN[19]. We then normalized each expression data set using quantile normalization. We computed gene-level expression for each sample by averaging expression values from probes mapping to the same genes while excluding individual probes that were promiscuously associated to more than one transcript. Datasets used to estimate technical bias are described in Supplementary Table 1. Dataset E-MTAB-62 [https://www.ebi.ac.uk/arrayexpress/experiments/E-MTAB-62/] was downloaded, processed, and annotated from Array Express (http://www.ebi.ac.uk/arrayexpress) using the ArrayExpress R package. Dataset GSE65133 [https://www.ncbi.nlm.nih.gov/geo/query/acc.cgi?acc=GSE65133] and its paired flow-cytometry data were directly downloaded from the CIBERSORT website (https://cibersort.stanford.edu). Paired flow-cytometry data for GSE59654 [https://www.ncbi.nlm.nih.gov/geo/query/acc.cgi?acc=GSE59654] was downloaded from ImmPort (https://immport.niaid.nih.gov/; study ID: SDY404). Data from the Stanford-Ellison cohort from years 2011 to 2013 was collected and processed as previously described[26]. Dataset E-MTAB-62 was downloaded, processed, and annotated from Array Express (http://www.ebi.ac.uk/arrayexpress) using the ArrayExpress R package.

**Creation of the immunoStates signature matrix.** We collected and processed 165 publicly available gene expression datasets comprising 6160 microarray samples profiling sorted human leukocytes. Datasets used to build immunoStates are described in Supplementary Data 1. All datasets were converted to gene-specific expression matrices using the original probe annotation files available from GEO and then combined into a single expression matrix using quantile normalization. Each sample was first annotated following experimental description from the original study, resulting in 47 different cell types. From these initial annotations, cells were grouped into more general categories in order to increase the number of studies and platforms represented in each cell type. We defined 20 different cell types: CD4+ T cells, CD8+ T cells, gamma-delta T cells, CD14+ monocytes, CD16+ monocytes, macrophages M0, macrophages M1, macrophages M2, CD56-high natural killer cells, CD56-dim natural killer cells, naïve B cells, memory B cells, plasma cells, myeloid dendritic cells, plasmacytoid dendritic cells, hematopoietic progenitors, MAST cells, neutrophils, eosinophils, and basophils. We then grouped cell types using manually defined lineages according to their biological similarity: T cells, monocytes, macrophages, natural killer cells, B cells, myeloid dendritic cells, plasmacytoid dendritic cells, hematopoietic progenitors, and granulocytes (see Supplementary Data 1 for annotation details). For each cell type within each lineage, we computed Hedge's g effect sizes to determine differential expression comparing a given cell type (cases) against all remaining cells within that lineage (controls). We applied a correction for small sample size bias to Hedge's g as needed. For each gene g and cell type i within lineage l, we computed an effect size as

$$\Delta E_{gil} = \min_j \left( E_{gil} - E_{gjl} \right) \quad (1)$$

where j is any cell type within lineage l such that $j \neq i$. A high effect size indicates a strong separation between our target cell type i and its closest cell type j.

We then ranked genes in decreasing order according to their effect sizes, and performed a step-wise search to identify the smallest gene signature able to accurately classify cell type i from every other cell type j in l. We first estimated the classification accuracy of the first gene of the list by estimating the area under receiver operating characteristic (AUROC) curve. Next, we incrementally added one gene at a time following their ranking and recomputed the AUROC corresponding to the new gene set. We repeated this process until we identified the minimal gene set that produced an AUROC proximal to the maximum (with $\varepsilon = 0.005$), requiring a minimum of 5 genes per signature. We performed this strategy on each cell type across each lineage and obtained a signature of 201 genes.

We then applied the same strategy to distinguished lineages from one another. We excluded all the signature genes that were used to separate cell types due to their confounding contribution in separating lineages. We performed the same gene selection strategy described above and obtained a second gene set of 116 genes. Together, these sets make the 317 genes that form immunoStates. To build the basis matrix with expression values, we computed the mean expression for every gene in every cell type (317 genes by 20 cell types) from the quantile normalized expression matrix (Supplementary Data 2). We compared our gene-set selection strategy with greedy forward-search and ranking by fold-change. We found using Hedge's g with our selection strategy to be more accurate in distinguishing cell types. All code was written and run using the R programming language.

**Cell-mixture deconvolution.** We performed deconvolution with support vector regression using the CIBERSORT algorithm (v1.03)[7]. We implemented linear model, quadratic programming, and robust regression methods using existing R programs and packages (lm, quadprog, and MASS)[3,4,7]. We used PERT as is from its source code in Octave[21]. We replicated all the pre-processing steps that CIBESORT performed on both the expression sample and basis matrix (quantile normalization and re-scaling of the matrix) in order not to be confounded when we

compared different methods. We assessed the effect of re-scaling for the linear model and robust regression, two methods that did not require re-scaling in their original implementation, by computing the pairwise correlation of estimated cell proportions with and without re-scaling (Supplementary Fig. 2). We observed high correlations for both methods, indicating that the effect of re-scaling was negligible. We therefore chose to maintain the same preprocessing strategy across all methods. We downloaded the LM22 basis matrix from the CIBERSORT website (https://cibersort.stanford.edu) and the IRIS matrix from the CellMix R package. Statistical comparisons were performed using the Wilcoxon's rank sum test. Analysis and plots were generated using the R programming language.

**Analysis of platform-dependent technical bias in deconvolution.** To quantify the extent of platform-dependent technical bias in deconvolution, we analyzed a collection of 1071 microarray samples profiling human PBMCs in diseased and healthy individuals profiled on platforms GPL96, GPL571, GPL570, GPL5715, GPL6244, GPL6104, GPL6947, GPL6102, and GPL6480 (Supplementary Table 1). We measured deconvolution performance using the goodness of fit score[7]. Briefly, for a given basis matrix **M** and a mixture sample gene expression vector **s**, deconvolution estimates the known cell proportion vector **p** such that

$$\mathbf{s} \alpha \mathbf{M} \times \mathbf{p} \quad (2)$$

Goodness of fit of the basis matrix **M** is defined as the Pearson correlation coefficient between **s** and the reconstituted expression vector **ŝ** defined as **M** × **p**. This value is indicative of how well a particular basis matrix **M** fits **s**. Tests comparing goodness of fits of individual platform/manufacturer were performed for each method using the Wilcoxon's rank sum test and were then integrated into a final p-value using Fisher's log sum rule. We calculated heterogeneity for a given matrix by computing the median absolute deviation (MAD) of the median goodness of fit across every platform and deconvolution method. We chose MAD because of its robustness to outliers in estimating the heterogeneity of the distribution of interest. To estimate significance of our MAD scores, we generated a background distribution of MAD scores for each platform, which represent an expected distribution of homogenous MAD values, and then performed a Z-test, asking whether the observed MAD was significantly higher than expected. All analysis and plots were generated using the R programming language.

**Analysis of the effect of disease state on deconvolution accuracy.** To estimate the effect of disease to sample deconvolution, we analyzed dataset E-MTAB-62 [https://www.ebi.ac.uk/arrayexpress/experiments/E-MTAB-62/] which contains samples profiled on Affymetrix HG-U133A arrays (GPL96). For the purpose of this analysis, we excluded all cell line samples and removed samples that had been used to generate immunoStates. Briefly, we deconvolved each sample across every matrix and method combination, and estimated a p-value indicative of the significance of its goodness of fit[7]. We then grouped all samples based on whether they originated from a healthy or a disease affected donor. Within each group, we compared the significance of goodness of fit between samples originating from blood and those originating from solid-tissue biopsies. Using the p-values as scores, we measured their accuracy in distinguishing blood-derived samples from solid tissue biopsies by computing an area under the receiver operating characteristic curve (AUROC), as previously described[7], for all combinations of basis matrices and deconvolution methods. Analysis and visualization was performed using the R programming language.

**Code availability.** Code for all the analysis can be found at https://khatrilab.stanford.edu/immunostates. All analyses and visualization were performed using the R programming language.

## Data availability

The data that support the findings of this study is either publicly available from GEO and Array Express, available from the authors of the original study, or provided as an archive file as indicated in the methods section, and can be provided by the author upon reasonable request.

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

## Acknowledgements

P.K. is funded by Bill & Melinda Gates Foundation, Vir Bio, and National Institute of Allergy and Infectious Diseases grants 1U19AI109662, U19AI057229, and RO1 AI125197 outside of the submitted work. F.V. is funded by NIH K12 5K12HL120001 fellowship.

## Author contributions

P.K. and F.V. conceived and conceptualized the study. F.V. performed analysis. A.T., S.L. and W.H. contributed to analysis and code. F.V., A.T., S.S., T.D.A., E.B., M.Alsup and M. Alonso collected and annotated public data. M.D. and E.E. provided data. F.V. and P.K. interpreted the results and wrote the manuscript.

## Additional information

**Competing interests:** F.V. and P.K. declare the following competing interests: immunoStates has been non-exclusively licensed by Vir Bio and have received licensing fees. The remaining authors declare no competing interests.

