## [Peer Review File · Nature Communications]

Reviewers' comments:

Reviewer #1 (Remarks to the Author):

Vallania et al. derived a novel basis matrix (immunoStates) for use in computational deconvolution of gene expression from mixtures of blood cell types. They claim that their method leverages heterogeneity across platforms and disease stages, resulting in a reduction in biases observed in deconvolution using existing blood basis matrices and providing improved deconvolution accuracy.

The problem of computationally deconvolving gene expression and cell type proportion variation in heterogeneous samples is an important one and blood is the most commonly targeted tissue for computational deconvolution, because cellular composition can vary substantially and can have clinically significant implications. It is reasonable to investigate the effect of using a broader range of samples in order to extract signature matrices. However, I was not convinced that the authors' description of this as leveraging heterogeneity to overcome platform or state-specific biases is entirely apt. Because the dataset they used to extract the signatures spans multiple different platforms, the signature genes they identify should naturally be more representative of genes that robustly behave as signatures across diverse datasets. Regardless of how they are described (and that is not a major concern) the claims appear to be supported by the results presented. Crucially, immoStates consistently outperformed two existing basis matrices, regardless of the deconvolution method used, on the key problem of recovering cell type proportions from mixed samples (for which the cell proportions had been experimentally determined).

In addition to providing a basis matrix that should lead to improved deconvolution of gene expression data from blood this manuscript also highlights the impact of the basis matrix on the accuracy of deconvolution and that performance may be more consistent across methods than between basis matrices. I believe this to be a novel result that has the capacity to influence the field. However, it was not true to the same extent for cell proportion estimation using the other basis matrices (particularly IRIS). This may need to be acknowledged more clearly in the manuscript and consideration given to an explanation of the sensitivity to choice of deconvolution method seen for one of the basis matrices (but not for immunoStates).

It was not clear to me that the authors' method for determining differential effect sizes takes into account the potential impact of differences in the number of samples for each cell type. Can the authors comment on whether their method is affected by this? Hedges g is also considered a biased estimator (for which a correction is available). Is it necessary to account for this bias or do they consider their method a heuristic for which the bias correction is not necessary? The choice of method to rank genes is not the only one and possibly not the obvious one. Several other methods have been applied in the past to identify signature genes. Have the authors compared to results obtained with different methods? Is the advantage of immoStates that it is derived from a more diverse set of input samples, or do the authors also have evidence to support the optimality of choice of method to derive the basis matrix?

Minor

Number the pages and equations for ease of review.

There was no mention in the manuscript of the role of single cell transcriptomics in defining cell subtypes as well as informing the design of basis matrices for future computational deconvolution methods. Could this be addressed in the Discussion?

"We then deconvolved the technical bias evaluation cohort..." the cohorts should be introduced.

'were converted in gene-specific expression matrices' should this 'in' be 'to'?

'in order to increase number of studies' missing article before 'number'

Typo in Methods "For a given a basis matrix"

Should the 'proportional to' sign in the equation giving the objective of deconvolution be \sim (i.e. approximated by).

'as the Pearson's correlation' should be 'as the Pearson correlation'

'Analysis of disease effect to deconvolution' This heading should be rephrased for clarity (at least 'to' should be 'on'). Occurs again in the first line of this section.

'...samples significantly reduce the biological bias and increase deconvolution accuracy' reduce -> reduces; increase -> increases (for concordance)

'high correlated cell proportion' -> 'highly correlated cell proportion'

'the reduction biological' missing 'in'

Reviewer #2 (Remarks to the Author):

Manuscript Summary

The authors developed an immune signature gene set - "ImmunoStates" - from a large collection of publicly available microarray data. They compared ImmunoStates with two other signature sets: IRIS and LM22. The authors found that the performance of their signature is superior to the two other signatures independent of the underlying deconvolution models. Overall, the experiments are interesting, but needs stronger data to support their claims.

Major Concerns:

1. The author used the goodness of fit, which is the Pearson's correlation coefficient between mixtures and reconstituted mixtures derived from deconvolution analysis, to evaluate the performance of different algorithms. I want to point out this metric is not appropriate to evaluate the performance of the deconvolution algorithms. To evaluate the performance of the algorithms, especially in the case when not enough cell counting data is available, pseudo-mixtures should be generated and estimated frequency and signature profiles (not only genes in the basis matrix, but for all genes in the mixture profile) should be compared to the ground truth.
2. For figure 1, even ImmunoStates showed some platform-based biases. Both IRIS and LM22 have been constructed from the gene expression profiles derived from the Affymetrix U133A, but IRIS performs best for datasets derived from the Illumina platform. The better performance of ImmunoStates is derived from the facts that it was constructed from larger datasets. Similar phenomenon can be seen from xCell paper too, where basis matrix has been constructed from both array-based platforms and sequencing platforms. I suggest author add the xCell along with other basis matrices for comparison. Still the observed good performance of ImmunoState is based on the "goodness of fit", as I have stated earlier, this is not a direct way to assess the performance of deconvolution algorithms.
3. The author observed limited difference between different methods. However, very limited

number of deconvolution methods have been tested. I would recommend author to add more deconvolution algorithms for comparison: LLSR, PERT, MMAD, ssNMF and DSA. PERT assumes the difference between the basis matrix expression and expression of certain cell type in the mixture profile so that the deconvolution is more feasible for the disease sample. It would be necessary to include these methods to clarify author's strong conclusion that no differences were observed among main deconvolution methods in this Manuscript.

4. Hedge's g score is biased upwards for small samples (< 50), the author should clarify the impact and try to use a modified version of the Hedge's g score to correct the bias.

5. In Figure 2, It was not clear how the AUC was generated. There are many confounding factors between the healthy and the "disease" population. It is not immediately clear whether the difference between these signatures is merely a reflection of the differences between these populations.

6. The result of Figure 3 is obvious and trivial. The basis matrix should have more impact on the deconvolution results compared to the methods.

Minor Concerns:

1. CIBERSORT performed the rescaling because support vector regression applies for the data with standard normal distribution. So, I suggest the author only keeps the rescaling strategy for CIBERSORT protocol, since this rescaling transformation is not fair for other methods.

2. The authors claimed that the code "Code for all the analysis can be found at <https://khatrilab.stanford.edu/immunoStates>." However, the codeset only contains the visualization, but not the analysis part of the project.

3. The sources where these signatures were generated are not clear. According to the authors, "Datasets used to build ImmunoStates are described in Supplementary Table 3." However, there is no such information in their Supp. Table 3. Only an ImmunoStates blood cell matrix was presented in Table 3.

4. In Page 7, the authors state "Datasets used to estimate accuracy by comparing with measured cell counts are described in Supplementary Table 2", However, when I looked into this table, it states "Datasets used to generate immunoStates". If both statement are true, you basically generated and evaluated your signature using the same set of data. This is a serious concern for overfitting for results in Figure 4.

5. In the Methods section, the math equation should be typed using proper math notes. S "alpha" M times P is odd.

Reviewer #3 (Remarks to the Author):

This work investigates the potential of deconvolving the cell type contributions of bulk transcriptomics measurements. Specifically, the authors study the biases introduced by type of sample, i.e. originating from diseased/healthy patients, and introduced by type of measurement platform. The authors further present a new single basis matrix for deconvolution across all these types of deconvolution situations.

It is not surprising that such biases exist and turn out to be pronounced. The main contribution of this work is making these biases explicit by analyzing an unprecedented bulk of transcriptomic data for deconvolution. It is not surprising that existing approaches, not being trained on all the data utilized in this study underperform.

However, the solution offered by the authors is likely to suffer from the same pitfalls as previous solutions since it builds on a single, one-fits-it-all basis matrix. Information about measurement

platform and sample type are known before the deconvolution analysis. Why not simply providing a specific basis matrix for each of these situations? Specifically, why not first defining a sensible set of deconvolution situations (by platform and sample type) and fitting a basis matrix for each of these situations. For a specific application, the user would have to look up its (possibly closest) deconvolution situation and deconvolve using the appropriate basis matrix.

Reviewer #1 (Remarks to the Author):

- *Vallania et al. derived a novel basis matrix (immunoStates) for use in computational deconvolution of gene expression from mixtures of blood cell types. They claim that their method leverages heterogeneity across platforms and disease stages, resulting in a reduction in biases observed in deconvolution using existing blood basis matrices and providing improved deconvolution accuracy.*
- *The problem of computationally deconvolving gene expression and cell type proportion variation in heterogeneous samples is an important one and blood is the most commonly targeted tissue for computational deconvolution, because cellular composition can vary substantially and can have clinically significant implications. It is reasonable to investigate the effect of using a broader range of samples in order to extract signature matrices. However, I was not convinced that the authors' description of this as leveraging heterogeneity to overcome platform or state-specific biases is entirely apt. Because the dataset they used to extract the signatures spans multiple different platforms, the signature genes they identify should naturally be more representative of genes that robustly behave as signatures across diverse datasets. Regardless of how they are described (and that is not a major concern) the claims appear to be supported by the results presented. Crucially, immoStates consistently outperformed two existing basis matrices, regardless of the deconvolution method used, on the key problem of recovering cell type proportions from mixed samples (for which the cell proportions had been experimentally determined).*

We thank the reviewer for the supportive comments. The reviewer is correct in pointing out that the use of multiple different platforms allowed us to identify more robust representative gene signatures for each cell type by accounting for technical variability between these platforms. However, we would like to point out that our use of the term “heterogeneity” also includes accounting for biological heterogeneity. Both IRIS and LM22 only used healthy control samples for deriving gene signatures for each cell type. In contrast, we also included samples from various diseases, with or without treatment, and with or without stimulation to represent biological heterogeneity in our data used for identifying cell-type specific genes. Our rationale was based on the observation in cytometry (FACS or CyTOF), where a cell surface marker is used as an immutable identifier of a cell type. For instance, CD14, CD56, and CD19 almost always represent monocytes, NK cells, and B cells, respectively, regardless of whether a sample comes from a healthy individual or from a patient with a disease. Similarly, our goal in this manuscript was to identify robust gene signatures that represent presence of a cell type with very high confidence. Traditionally, these factors (disease status, treatment, stimulation) are considered confounding factors (as mentioned by Reviewer 2 below) due to underlying biological heterogeneity. Hence, our use of the term “heterogeneity” is to ensure that both biological and technical heterogeneity are accounted for in the basis matrix. We have maintained the use of the term, but remain open to changing it to an appropriate term suggested by the reviewer and/or the editor.

- *In addition to providing a basis matrix that should lead to improved deconvolution of gene expression data from blood this manuscript also highlights the impact of the basis matrix on the accuracy of deconvolution and that performance may be more consistent across methods than between basis matrices. I believe this to be a novel result that has the capacity to influence the field. However, it was not true to the same extent for cell proportion estimation using the other basis matrices (particularly IRIS). This may need to be acknowledged more clearly in the manuscript and consideration given to an explanation of the sensitivity to choice of deconvolution method seen for one of the basis matrices (but not for immunoStates).*

We thank the reviewer for this constructive suggestion. We have modified the Discussion section to explicitly explain that choice of deconvolution method for a basis matrix is an important consideration when it is not created using heterogeneous data similar to immunoStates. Following the reviewer's suggestion, we have modified the Discussion section as follows:

“Improvements in methods are usually demonstrated in particular contexts, such as estimating proportions in the presence of background tissue⁷ or robustness to expression changes due to treatments²¹. The design and execution of these studies typically involve the application of the same basis matrix, and results in improvements in accuracy provided by the new method for the dataset that represent the use case of interest. While the choice of method is therefore important in such specific cases, our results strongly support the argument that improvements on the basis matrix improve accuracy across all methods and datasets.”

- *It was not clear to me that the authors' method for determining differential effect sizes takes into account the potential impact of differences in the number of samples for each cell type. Can the authors comment on whether their method is affected by this? Hedges g is also considered a biased estimator (for which a correction is available). Is it necessary to account for this bias or do they consider their method a heuristic for which the bias correction is not necessary? The choice of method to rank genes is not the only one and possibly not the obvious one. Several other methods have been applied in the past to identify signature genes. Have the authors compared to results obtained with different methods? Is the advantage of immoStates that it is derived from a more diverse set of input samples, or do the authors also have evidence to support the optimality of choice of method to derive the basis matrix?*

The reviewer brings up multiple points. First, we corrected for the small sample bias as needed when computing Hedges' g. We have modified the Methods section to explicitly state it as follow:

“We applied a correction for small sample size bias to Hedge's g as needed.”

Second, indeed many different methods can be used to rank genes. We chose Hedge's g because of our experience in using it to generate and validate robust and consistent gene expression signatures (Khatri *et al. J. Exp Med* 2013, Sweeney *et al. Science Trans Med* 2015, Andres-Terres *et al. Immunity* 2015, Sweeney *et al. Science Trans Med* 2016, Sweeney *et al. Lancet Resp Med* 2016, Chowdhury *et al. Nature* 2018). Indeed, during the development of immunoStates, we tried several methods to select cell-type specific genes, which include greedy forward-search and ranking by fold-change. For each method, we selected a set of genes for each immune cell type using the discovery cohorts, and evaluated accuracy of these genes sets in identifying the correct cell type in the validation cohorts. Across these methods we found that gene-set selection by Hedges' g as described in the manuscript performed the best in distinguishing immune cell types. We have included the following in the Methods section:

“We compared our gene-set selection strategy with greedy forward-search and ranking by fold-change. We found using Hedge's g with our selection strategy to be more accurate in distinguishing cell types (data not shown).”

Third, when evaluating different gene sets using different selection strategies, we found that the accuracy of a basis matrix is strongly dependent on its set of signature genes rather than the expression values of the gene themselves. We generated a basis matrix using the genes from the LM22 matrix and the expression values from the datasets used to construct immunoStates. We found that the pattern of platform bias mirrored that of LM22. This result also further supported our choice of using the Hedges' g to select robust cell-type specific genes as it has been shown to do for disease-specific genes.

In response to the reviewer's comment, we have modified the Methods section to mention other methods we evaluated. We have also included a new supplementary figure (**Supplementary Figure 4** in the revised manuscript) to illustrate that the set of genes matter more than expression values themselves. We have modified the Results section to include the following:

“Arguably, higher goodness of fit of immunoStates could be due to higher amount of data used to create it. It is possible that if the same amount of data were used to create one of the existing basis matrices, it would have higher goodness of fit as well. We investigated this argument by modifying LM22 such that it contained the same genes, but their expression values were computed using the data sets used for creating immunoStates. We found that despite increasing the amount of data used to estimate expression values for LM22 genes, it continued to have platform bias as before with significant heterogeneity in goodness of fit (MAD = 0.05, $p = 0.05$, **Supplementary Figure 4**). These results suggest that better estimation of expression values for genes in a basis matrix using large amount of data is not sufficient to increase deconvolution accuracy. Further, these results strongly suggest that selection of genes in a basis matrix from biologically and technologically heterogeneous data is more important in reducing bias. Together, our results demonstrate that a basis matrix created using heterogeneous data from multiple platforms reduces technical bias.”

Minor

- Number the pages and equations for ease of review.

We have inserted page and equation numbers

- There was no mention in the manuscript of the role of single cell transcriptomics in defining cell subtypes as well as informing the design of basis matrices for future computational deconvolution methods. Could this be addressed in the Discussion?

This is a point well taken. We have revised the Discussion section at two places as follows:

“We expect that continued accumulation of additional sorted-cell datasets in public repositories over time will increase both the accuracy and the breadth of future basis matrices. Furthermore, increased availability of single cell RNAseq data will further facilitate creation of better and more refined basis matrices.”

and

“Third, arguably, all basis matrices require *a priori* knowledge of the populations within the sample of interest. However, current cytometry-based methods also require *a priori* selection of markers. We expect these limitations will be overcome as more single-cell transcriptomic data become available. These data will allow for the discovery of previously unknown cell subsets that can be included in a basis matrix. These data will also increase heterogeneity and the sample size, which based on our analysis presented here, will further improve accuracy of deconvolution.”

- *“We then deconvolved the technical bias evaluation cohort...” the cohorts should be introduced.*

We have modified the Results section to introduce the evaluation cohort as follows:

“We deconvolved 17 independent datasets consisting of 1071 whole transcriptome profiles of human peripheral blood mononuclear cells (PBMCs) measured across eight microarray platforms from two different manufacturers (see Methods and **Supplementary Table 1**) using both basis matrices. We define this set as a “technical bias evaluation cohort”.”

- *‘were converted in gene-specific expression matrices’ should this ‘in’ be ‘to’?*

Thank you for pointing out the error. We have fixed the typo.

- *‘in order to increase number of studies’ missing article before ‘number’*

We have rephrased this as “in order to increase the number of studies”.

- *Typo in Methods “For a given a basis matrix”*

We have removed the extra “a” before “basis matrix”.

- *Should the ‘proportional to’ sign in the equation giving the objective of deconvolution be \sim (i.e. approximated by).*

We have replaced “proportional to” sign with “approximation” sign.

- *‘as the Pearson’s correlation’ should be ‘as the Pearson correlation’*

We have corrected the typo.

- *‘Analysis of disease effect to deconvolution’ This heading should be rephrased for clarity (at least ‘to’ should be ‘on’). Occurs again in the first line of this section.*

Thank you for the suggestion. We have modified the manuscript to say “on”.

- *‘...samples significantly reduce the biological bias and increase deconvolution accuracy’ reduce -> reduces; increase -> increases (for concordance)*

We have corrected these errors.

- *‘high correlated cell proportion’ -> ‘highly correlated cell proportion’*

We have corrected the typo.

- *‘the reduction biological’ missing ‘in’*

We have inserted “in”.

Reviewer #2 (Remarks to the Author):

Manuscript Summary

The authors developed an immune signature gene set - "ImmunoStates" - from a large collection of publicly available microarray data. They compared ImmunoStates with two other signature sets: IRIS and LM22. The authors found that the performance of their signature is superior to the two other signatures independent of the underlying deconvolution models. Overall, the experiments are interesting, but needs stronger data to support their claims.

Major Concerns:

- The author used the goodness of fit, which is the Pearson's correlation coefficient between mixtures and reconstituted mixtures derived from deconvolution analysis, to evaluate the performance of different algorithms. I want to point out this metric is not appropriate to evaluate the performance of the deconvolution algorithms. To evaluate the performance of the algorithms, especially in the case when not enough cell counting data is available, pseudo-mixtures should be generated and estimated frequency and signature profiles (not only genes in the basis matrix, but for all genes in the mixture profile) should be compared to the ground truth.

We are in complete agreement with the reviewer that using cell count data, especially from high-resolution technologies such as FACS and CyTOF, is the most appropriate way to evaluate the performance of deconvolution algorithms. We further argue that it is a higher bar for estimating accuracy of a deconvolution method than creating a pseudo-mixture that may or may not represent real biology. Therefore, as shown in **Figure 4** and **Supplementary Figures 7-14**, we evaluated deconvolution accuracy of each of the 3 basis matrices across 5 deconvolution methods using 402 human blood samples from 5 independent datasets. These datasets are published from independent groups that generated transcriptome data using different microarrays (Affymetrix and Illumina) and also performed cellular profiling to identify cellular proportions in each sample. In these real-world data, we showed that irrespective of the method used, immunoStates had higher positive correlations between estimated and measured cellular proportions from these samples, just as the reviewer suggested.

We realized we were not clear in our manuscript about this comparison. In response to the reviewer's comment we have added the following statement to make this comparison explicit:

"Despite demonstrated utility of goodness of fit in evaluating accuracy of deconvolution⁷, cell count data from high-resolution technologies such as FACS is the most appropriate way to evaluate the performance of deconvolution algorithms. Therefore, we explored ..."

Most importantly, we evaluated ability of basis matrices (immunoStates and LM22) to identify hitherto unknown changes in cell proportions between healthy controls and those with latent mycobacterium (*Mtb*) infection (LTBI). In an independent manuscript – now accepted for publication in Nature – we deconvolved whole blood transcriptome data from 12 datasets composed of 1276 samples using immunoStates, and found that NK cell proportions are higher in LTBI patients than healthy controls ($P = 2.85e-05$, FDR = 0.018%). In comparison, LM22 did not identify the increase in NK cell proportions in LTBI subjects compared to healthy controls ($P = 0.157$, FDR = 25.8%). We did not compare IRIS as our analysis in the current manuscript demonstrated it had the lowest accuracy among the 3 basis matrices. Next, we validated increased NK cells in LTBI patients compared to healthy controls using CyTOF and flow cytometry in an independent cohort of adolescents from South Africa (N=52, 26 healthy controls, 26 LTBI; **Figure R1A-C** below).

Similarly, in the same analysis we found that according to immunoStates, T cell proportions did not change between healthy controls and LTBI subjects ($P = 0.71$), but changed significantly according to LM22 ($P = 6.18e-03$, FDR = 5.8%). Cellular phenotyping of PBMC samples from adolescents in South Africa using CyTOF or FACS

showed that none of the T cell subsets changed in LTBI subjects compared to healthy controls (Figure R1D-F below).

These results clearly demonstrate that immunoStates has substantially increased deconvolution accuracy that identifies hitherto unknown biology that is independently validated using CyTOF and FACS. These results are not included in the revised manuscript, as they are part of an already accepted independent manuscript. These results are the most stringent independent evaluation and validation of immunoStates and its application to identify hitherto unknown biology that the other basis matrices missed.

Figure R1. CyTOF and FACS in an independent cohort validate changes predicted immunoStates, but not by LM22. We performed deconvolution of whole blood transcriptome data from children and adults using immunoStates or LM22, and compared with cellular proportions measured using CyTOF or FACS in an independent cohort of PBMC samples from adolescents from South Africa. Deconvolution using (A) immunoStates predicted increase in NK cell proportions in LTBI subjects, but (B) LM22 predicted no change. (C) Cellular phenotyping of PBMC samples from adolescents in South Africa showed increased NK cells in LTBI subjects, which validated immunoStates results. Deconvolution using (D) immunoStates predicted no change in T cell proportions in LTBI subjects, but (E) LM22 predicted no change. (F) Cellular phenotyping of PBMC samples from adolescents in South Africa showed no change in any subject of T cells in LTBI subjects, which validated immunoStates results. Positive and negative effect size indicate higher and lower proportion, respectively, in LTBI patients compared to healthy controls.

Given our use of more stringent evaluation criteria than suggested by the reviewer, and demonstrated increased accuracy of deconvolution using the same basis matrices in another manuscript, we have refrained from creating pseudo-mixtures in the revised manuscript.

Finally, we are confused by the reviewer's comment about "signature profiles for not only genes in the basis matrix, but for all genes in the mixture profile". As we expect the reviewer is aware, cell mixture deconvolution is broadly divided into two types of estimations. First, given a basis matrix, estimating proportions of various cell types within a sample, and second, given cellular proportions, estimating cell-type specific expression profiles that estimates expression of each gene in each cell type. Is the reviewer referring to the second type of deconvolution? If yes, we note that our manuscript is only focused on the first type of deconvolution (estimating cellular proportions), and estimating expression of each gene in each cell type is out of scope of our manuscript. For further details, we reference a comprehensive review by Shen-Orr and Gaujoux, *Current Opinion in Immunology* 2013, 25(5):571-578, that discusses various methods spanning both types of deconvolution approaches.

- 2. *For figure 1, even ImmunoStates showed some platform-based biases. Both IRIS and LM22 have been constructed from the gene expression profiles derived from the Affymetrix U133A, but IRIS performs best for datasets derived from the Illumina platform. The better performance of ImmunoStates is derived from the facts that it was constructed from larger datasets. Similar phenomenon can be seen from xCell paper too, where basis matrix has been constructed from both array-based platforms and sequencing platforms. I suggest author add the xCell along with other basis matrices for comparison. Still the observed good performance of ImmunoState is based on the "goodness of fit", as I have stated earlier, this is not a direct way to assess the performance of deconvolution algorithms.*

The reviewer brings up multiple points. First, the reviewer is correct that immunoStates still showed some platform biases. We have been very conscious to ensure that we never claimed that immunoStates *removed* all biases. We emphasized this issue by stating in the title of our manuscript that it "reduces biological and technical biases". In response to the reviewer's comment, we have included the following in the Discussion to state it explicitly:

"Second, we emphasize that immunoStates did not remove these biases entirely, but reduced them substantially, which in turn significantly improved accuracy of deconvolution. We expect that these biases will continue to reduce as more data for sorted cells becomes available and are used to update immunoStates in the future."

Next, we agree with the reviewer that the better performance of immunoStates is derived from the fact that it was constructed from larger datasets. The goal of our work is to highlight how incorporating large amounts of data improves performance of cell-mixture deconvolution by reducing biological and technical biases. Similar philosophy underlies the design of xCell. However, we would like to point out that xCell is not a basis matrix but a signature-based method. xCell is used to estimate which cell types are enriched – as represented by xCell scores – in a sample for a given set of genes. It does not estimate proportions of various cell types in a sample. xCell belongs to a different group of methods that are not comparable to deconvolution methods, and are not the focus of our work. Therefore, we have chosen not to include xCell in the revised manuscript.

Finally, with respect to the suitability of goodness of fit, we reiterate that goodness of fit is one of the ways we assessed performance of deconvolution algorithms. As we stated above, we additionally used more stringent criteria for assessing performance of these basis matrices and algorithms by comparing estimated and measured cellular proportions from 5 independent cohorts of >400 blood samples. Finally, we have demonstrated accuracy

of immunoStates in identifying hitherto unknown changes in cellular proportions in latent *Mtb* infection that other basis matrices failed to identify (please see **Figure R1** above).

- 3. The author observed limited difference between different methods. However, very limited number of deconvolution methods have been tested. I would recommend author to add more deconvolution algorithms for comparison: LLSR, PERT, MMAD, ssNMF and DSA. PERT assumes the difference between the basis matrix expression and expression of certain cell type in the mixture profile so that the deconvolution is more feasible for the disease sample. It would be necessary to include these methods to clarify author's strong conclusion that no differences were observed among main deconvolution methods in this Manuscript.

This is a point well taken. First, we note that LLSR is already included in our initial submission, labeled as Linear Model. We had considered DSA and ssNMF for comparison but excluded them because these methods rely on marker genes and do not use a basis matrix. We also excluded MMAD for our comparison as it has been shown to be more accurate as a marker gene approach than expression-based deconvolution due to the presence of scaling artifacts (Liebner *et al.* Bioinformatics 2013, pages 682-689). We observed significantly lower deconvolution accuracy using any of the 3 basis matrices with MMAD (**Figure R2**).

Finally, in response to the reviewer's comment, we have included PERT in the revised manuscript. As **Figure R2** above (and revised **Figure 1** in the manuscript) Our results remain unchanged when including PERT. All figures in the manuscript have been updated to include deconvolution results when using PERT.

- 4. Hedge's g score is biased upwards for small samples (< 50), the author should clarify the impact and try to use a modified version of the Hedge's g score to correct the bias.

The reviewer raises the same concern as Reviewer 1. We corrected for the small sample bias as needed when computing Hedges' g. We have clarified this point in the methods section.

- 5. In Figure 2, It was not clear how the AUC was generated. There are many confounding factors between the healthy and the "disease" population. It is not immediately clear whether the difference between these signatures is merely a reflection of the differences between these populations.

We apologize for the confusion. For comparison purposes, Figure 2 in our manuscript presents the results of an analysis that is exactly the same as described in Figure 1C by Newman *et al* Nature Methods 2015 performed on the same dataset (E-MTAB-62). This dataset is a compendium of gene expression data profiled across a single Affymetrix platform (GPL96); hence, there is no technical variation, but contains biological heterogeneity as it integrates samples from multiple diseases, showing that our results are not driven by a single study. These samples consist of transcriptomes from blood and tissue biopsy samples from healthy and disease-affected individuals. Briefly, we deconvolved each sample, estimated the significance of the goodness of fit as previously described by Newman *et al* Nature Methods 2015, and used it as a score to distinguish blood-derived samples from biopsies for either healthy or disease-affected samples. We modified the Methods section to clarify this analysis as follows:

"To estimate the effect of disease to sample deconvolution, we analyzed dataset E-MTAB-62 which contains samples profiled on Affymetrix HG-U133A arrays (GPL96). For the purpose of this analysis, we excluded all cell line samples and removed samples that had been used to generate immunoStates. We performed this analysis as described previously⁷. Briefly, we deconvolved each sample across every matrix and method combination, and estimated a p-value indicative of the significance of its goodness of fit as previously described⁷. We then grouped all samples based on whether they originated from a healthy or a disease affected donor. Within each group, we compared the significance of goodness of fit between samples originating from blood and those originating from solid-tissue biopsies. Using the p-values as scores, we measured their accuracy in distinguishing blood-derived samples from solid tissue biopsies by computing an area under the receiver operating characteristic curve (AUROC), as previously described⁷, for all combinations of basis matrices and deconvolution methods. Analysis and visualization was performed using the R programming language. Code for all the analysis can be found at <https://khatrilab.stanford.edu/immunoStates>."

- 6. The result of Figure 3 is obvious and trivial. The basis matrix should have more impact on the deconvolution results compared to the methods.

We respectfully disagree with the reviewer that the results are obvious and trivial. As Reviewers 1 and 3 have pointed out in their comment, it is a novel finding with unprecedented amount of data, and will move the field forward. For more than 17 years — since deconvolution was first described in 2001 — the field has been mostly focused on developing novel methods instead of developing better basis matrix. This is evident from the fact that most of the existing basis matrices were created by using as much data as possible, while restricting either biological or technical heterogeneity or both. The fundamental focus in the field has been to develop a method to improve accuracy of deconvolution instead of a basis matrix, which as our results show, has more impact on the deconvolution. One of our goals in this manuscript is to explicitly point out this observation that has received virtually no attention in the literature. We provide a method to demonstrate how it can be done. Our hope, as Reviewer 1 pointed out, is that the field will take note of this and invest more efforts in developing matrices that

are more robust and continue to reduce methodological, technical, and biological biases to further improve cell mixture deconvolution that can in turn be used to leverage existing transcriptome data to identify hitherto unknown biology as we demonstrate in **Figure R1** above using results from our upcoming manuscript in Nature.

Minor Concerns:

- 1. CIBERSORT performed the rescaling because support vector regression applies for the data with standard normal distribution. So, I suggest the author only keeps the rescaling strategy for CIBERSORT protocol, since this rescaling transformation is not fair for other methods.

This is a point well taken. In principle, re-scaling can be unfair to deconvolution methods other than CIBERSORT. To test the effects of re-scaling, we compared results from linear model and robust regression with and without re-scaling where in principle re-scaling can be detrimental. We found that the results were highly correlated observed high correlations for both methods, indicating that the effect of re-scaling is negligible. In response to the reviewer's comment, we have included a new supplementary figure in the revised manuscript to show our comparative analysis, as well as a description in the methods section.

"Arguably, support vector regression re-scales expression data prior to deconvolution, which is not required for other methods such as linear model and robust regression. Rescaling gene expression data when using these methods could potentially reduce accuracy of these methods. Therefore, we compared estimated proportions for linear model and robust regression with and without rescaling gene expression data in the technical bias evaluation cohort using IRIS and LM22 as basis matrices. We found there was very high correlation in estimated cellular proportions suggesting rescaling did not adversely affect linear model and robust regression (**Supplementary Figure 2**). Based on these results, we chose to maintain the uniform preprocessing strategy across all methods for the rest of the manuscript."

- 2. The authors claimed that the code "Code for all the analysis can be found at <https://khatrilab.stanford.edu/immunoStates>." However, the codeset only contains the visualization, but not the analysis part of the project.

We have included all the results of deconvolution for all expression datasets across all matrices and methods. We have also ensured that all code performs all analyses described in each figure, and produces the figures in the manuscript. We have divided our code into two parts: (1) perform all analyses and save results as an R object, and (2) use R object from the first part and generates figures.

- 3. The sources where these signatures were generated are not clear. According to the authors, "Datasets used to build ImmunoStates are described in Supplementary Table 3." However, there is no such information in their Supplementary Table 3. Only an ImmunoStates blood cell matrix was presented in Table 3.

We sincerely apologize for the mistake. We incorrectly referenced Supplementary Table 3 instead of Supplementary Table 2 in the manuscript. We have fixed the error. The correct statement now in the manuscript is:

"Datasets used to build immunoStates are described in **Supplementary Table 2**."

- 4. In Page 7, the authors state "Datasets used to estimate accuracy by comparing with measured cell counts are described in Supplementary Table 2", However, when I looked into this table, it states "Datasets used to generate

immunoStates". If both statements are true, you basically generated and evaluated your signature using the same set of data. This is a serious concern for overfitting for results in Figure 4.

We sincerely apologize for the mistake. We incorrectly referenced Supplementary Table 2. There is no table for these datasets as they are described inline in the manuscript as follows:

"These datasets were generated using Illumina HT12 V4.0 or Affymetrix Primeview microarrays as follows: (1) two independent datasets consisting of 176 healthy human PBMC samples profiled using Illumina HT-12 V4.0 arrays paired with flow-cytometry data (GSE65133, GSE59654)^{7,25}, and (2) a whole blood dataset of 226 healthy samples, a subset of which were profiled over three consecutive years using Affymetrix PrimeView arrays (see Methods)."

- 5. In the Methods section, the math equation should be typed using proper math notes. S^{α} M times P is odd.

We have modified the manuscript following the reviewer's suggestion.

Reviewer #3 (Remarks to the Author):

- *This work investigates the potential of deconvolving the cell type contributions of bulk transcriptomics measurements. Specifically, the authors study the biases introduced by type of sample, i.e. originating from diseased/healthy patients, and introduced by type of measurement platform. The authors further present a new single basis matrix for deconvolution across all these types of deconvolution situations.*
- *It is not surprising that such biases exist and turn out to be pronounced. The main contribution of this work is making these biases explicit by analyzing an unprecedented bulk of transcriptomic data for deconvolution. It is not surprising that existing approaches, not being trained on all the data utilized in this study underperform.*

We thank the reviewer for recognizing the main contribution of our work as unprecedented.

- *However, the solution offered by the authors is likely to suffer from the same pitfalls as previous solutions since it builds on a single, one-fits-it-all basis matrix. Information about measurement platform and sample type are known before the deconvolution analysis. Why not simply providing a specific basis matrix for each of these situations? Specifically, why not first defining a sensible set of deconvolution situations (by platform and sample type) and fitting a basis matrix for each of these situations. For a specific application, the user would have to look up its (possibly closest) deconvolution situation and deconvolve using the appropriate basis matrix.*

This is a point well taken. It is possible that having a specific basis matrix for different situations may be better than one-size-fits-all solution. However, we chose one-size-fits-most approach due to a number of practical considerations described below.

First, we believe that instead of replacing cytometry-based experimental approaches such as FACS or CyTOF, statistical deconvolution of mixed-tissue expression profiles will complement these technologies. Current cytometry approaches are limited in the number of variables they can profile simultaneously due to the number of channels available for measurements. As we demonstrated in response to a comment by Reviewer 2, a one-size-fits-most approach such as *immunoStates* is simpler from a practical standpoint as a researcher could directly apply without having to select the matrix most appropriate for their dataset of interest. The researchers can then design a panel of cell type markers for a mass cytometry experiment that is focused on the cell types identified through statistical deconvolution of transcriptome data. Such a strategy would reduce 'wasting' of

channels by allowing the researchers to eliminate markers unlikely to show any differences, and better utilize the available channels by including markers that allow better phenotyping.

Second, the amount of sorted-cell transcriptome data available for different platforms can be extremely limited, making it nearly impossible to build a robust basis matrix for each available platform. We have previously demonstrated that incorporating multiple heterogeneous data sets is the key in identifying robust gene expression signatures (Khatri *et al* 2013 *J. Exp Med.*, Sweeney *et al* 2015 *Science Trans. Med*, Andres-Terres *et al* *Immunity* 2015, Sweeney *et al* *NAR* 2017). As we try to divide available data into specific platforms, they become very scarce such that even within a given microarray platform, they may not be representative of biological heterogeneity.

Finally, as we mentioned in response to a comment by Reviewer 1, we found that the choice of genes is more important than their expression value. We have included the results of this analysis as **Supplementary Figure 4** in the revised manuscript. Using data from across multiple datasets profiled using different platforms is likely to identify a robust set of cell type specific genes than limited amount of platform-specific data, as it may not be sufficient to build a robust platform-specific basis matrix. Furthermore, we found that, by incorporating sufficient heterogeneity, our matrix was accurate on expression data profiled using platforms that were not represented in training and RNA-seq. Results of this analysis have recently been accepted for publication on a separate paper (Chowdury RR *et al.* *Nature* 2018 *in press*). We thank the reviewer for the insightful comment. We have updated the Discussion section as follows in response to the reviewer's comment:

“Our results across multiple independent platforms also suggest that immunoStates may be a one-size-fits-most basis matrix readily applicable to future deconvolution methods that rely on the use of a basis matrix to estimate cell proportions. This strategy has a number of practical advantages. First, researchers could directly apply immunoStates without having to select a basis matrix most appropriate for their dataset of interest. Second, the application of a single basis matrix facilitates comparison and integration of cell proportions from multiple datasets across multiple platforms and different centers. Third, the amount of sorted-cell data available for different platforms can be extremely limited, compromising the robustness of basis matrices created for each available platform. Fourth, we believe that instead of replacing cytometry-based experimental approaches such as FACS or CyTOF, statistical deconvolution of mixed-tissue expression profiles will complement these technologies. Current cytometry approaches are limited in the number of variables they can profile simultaneously due to the number of channels available for measurements. a one-size-fits-most approach such as immunoStates is simpler from a practical standpoint as a researcher could directly apply without having to select the matrix most appropriate for their dataset of interest. The researchers can then design a panel of cell type markers for a mass cytometry experiment that is focused on the cell types identified through statistical deconvolution of transcriptome data. Such a strategy would reduce ‘wasting’ of channels by allowing the researchers to eliminate markers unlikely to show any differences, and better utilize the available channels by including markers that allow better phenotyping.”

Reviewers' comments:

Reviewer #1 (Remarks to the Author):

The authors have addressed all of the points raised in my review.

Reviewer #2 (Remarks to the Author):

The authors addressed most of my concerns.

One last concern is the open source of the analysis code. The authors still didn't provide the source code for their analysis... The end results were saved as RDS files, but the parameters and scripts are still missing... It should be deposited for reproducibility purposes.

Reviewer #3 (Remarks to the Author):

The authors do not address my only major concern about the proposed deconvolution approach with a single, one-fits-it-all basis matrix, that conceptually does not differ from the already available deconvolution approaches and will likely suffer from the same pitfalls as these.

The authors elaborate in more detail on three reasons why they stick to a single basis matrix solution and do not consider the suggested multi basis matrix solution that incorporates the a priori known information about measurement platform and sample type.

As the first reason the authors put forward the practical simplicity of the single matrix approach. However, I do not see a significant increase in complexity for the user for a multi matrix solution. The user would only have to additionally specify the type of transcriptomics platform that was used, the software would subsequently automatically choose the corresponding basis matrix and compute the deconvolution.

Further the authors claim that incorporating multiple datasets "is the key in identifying robust gene expression signatures" and list a few own previous publications to back up this claim. These publications report results on integrating multiple transcriptomic datasets in different contexts. However, none of these publications integrates these datasets to the end of cell type deconvolution, and therefore do not support the benefit of integrating multiple datasets for single matrix based cell type deconvolution.

As the third reason the authors put forward better generalization capability of their single matrix to analyze transcriptomic datasets from new platforms. This is a fair statement but not getting to the initial point that a multi-matrix solution could achieve superior performance on the platforms considered for deriving the basis matrix.

It would be straightforward to address this concern by demonstrating the performance of a multi-matrix solution for the data considered in this study and comparing it to the performance of the proposed single-matrix solution.

Reviewer #3 (Remarks to the Author):

- The authors do not address my only major concern about the proposed deconvolution approach with a single, one-fits-it-all basis matrix, that conceptually does not differ from the already available deconvolution approaches and will likely suffer from the same pitfalls as these. The authors elaborate in more detail on three reasons why they stick to a single basis matrix solution and do not consider the suggested multi basis matrix solution that incorporates the a priori known information about measurement platform and sample type.*

We respectfully disagree with the reviewer for the reasons described below.

- As the first reason the authors put forward the practical simplicity of the single matrix approach. However, I do not see a significant increase in complexity for the user for a multi matrix solution. The user would only have to additionally specify the type of transcriptomics platform that was used, the software would subsequently automatically choose the corresponding basis matrix and compute the deconvolution.*
- Further the authors claim that incorporating multiple datasets “is the key in identifying robust gene expression signatures” and list a few own previous publications to back up this claim. These publications report results on integrating multiple transcriptomic datasets in different contexts. However, none of these publications integrates these datasets to the end of cell type deconvolution, and therefore do not support the benefit of integrating multiple datasets for single matrix-based cell type deconvolution.*

In our previous response, we provided an example from our manuscript, accepted for publication in *Nature* (**Figure R1** provided below for convenience), that demonstrated immunoStates outperformed LM22. We apologize for not providing additional details at the time that we now provide below.

In this upcoming manuscript, we deconvolved 15 datasets using immunoStates that were profiled using 9 microarray platforms from 4 microarray manufacturers and RNA-seq (Table 1). These datasets profiled blood samples from children, adolescents, and adults from 9 countries. Despite the biological (age, countries, etc.) and technical (microarray, RNA-seq) heterogeneity observed in these datasets, we identified robust changes in NK cell proportions associated with different stages of *Mycobacterium tuberculosis* (*Mtb*) infection and protection to progression to active tuberculosis disease. This analysis is a strong evidence in favor of integrating multiple datasets for single matrix-based cell type deconvolution to identify hitherto unknown changes in cellular proportions.

When presenting **Figure R1**, we incorrectly assumed that the broader (and lengthy) discussion was out of scope in the response. We acknowledge that **Figure R1** only presented a subset of these data and analysis, and in the context of comparing immunoStates and LM22 without highlighting integration of multiple cohorts in the context of cell type deconvolution. In light of the reviewer’s comment, we are happy to provide the manuscript prior to its publication for the reviewer, although the manuscript is scheduled for online publication on August 22, 2018, and in print on August 30, 2018.

Microarray	GSE ID	GPL ID
Affymetrix	GSE41055	GPL5175
	GSE54992	GPL570
Agilent	GSE25534	GPL1708
	GSE28623	GPL4133
	GSE62147	GPL6480
Illumina	GSE37250	GPL10558
	GSE39939	GPL10558
	GSE39940	GPL10558
	GSE40553	GPL10558
	GSE56153	GPL6883
Phalanx	GSE116014	GPL10558
	GSE19491	GPL6947
	GSE62525	GPL16951
RNA-seq	Adolescent Cohort Catalysis Cohort	

Table 1. List of 12 tuberculosis datasets profiled using 9 microarray platforms from 4 microarray manufacturers. Two datasets were profiled using RNA-seq.

- As the third reason the authors put forward better generalization capability of their single matrix to analyze transcriptomic datasets from new platforms. This is a fair statement but not getting to the initial point that a multi-matrix solution could achieve superior performance on the platforms considered for deriving the basis matrix.
It would be straightforward to address this concern by demonstrating the performance of a multi-matrix solution for the data considered in this study and comparing it to the performance of the proposed single-matrix solution.

We thank the reviewer for this excellent helpful suggestion. Indeed, as the reviewer suggested, **Figure 2** in the manuscript already demonstrated superior performance of a single matrix solution (i.e., immunoStates) compared to the performance of the reviewer's proposed multi-matrix solution in the manuscript. Briefly, **Figure 2** compared performance of the three basis matrices in E-MTAB-62 that profiled >4,000 samples using Affymetrix, using different deconvolution methods. Although IRIS and LM22 were created using only sorted

immune cell expression data from Affymetrix, immunoStates outperformed both Affymetrix-specific basis matrices, irrespective of the deconvolution method used. Similar comparison is also present in **Figure 4**, where each of the Ellison cohorts were profiled using Affymetrix. As shown in **Figure 4**, immunoStates consistently outperformed both Affymetrix-specific basis matrices irrespective of the deconvolution method used.

Following the reviewer's suggestion, we created two Illumina-specific basis matrices. One of these two basis matrices is specific to GPL10558 as it was created using only sorted immune cell expression profiles using this microarray platform (8 datasets, 336 samples); the second Illumina-specific basis matrix was created using sorted immune cell expression profiles that were generated using any of the Illumina microarrays (28 datasets, 1,670 samples).

Using these two basis matrices and immunoStates, we deconvoluted GSE65133, which was profiled using Illumina microarray platform GPL10558 (**Figure R2**). As expected, expression data using GPL10558 is available for a limited number of immune cells. Hence, a GPL10558-specific basis matrix is missing a number of important cell types. Importantly, when we deconvoluted GSE65133 using GPL10558-specific basis matrix, there was inverse correlation between estimated and measured cellular proportions, irrespective of the method used (first column in **Figure R2**).

Data for more immune cell types were available when using all microarrays from Illumina, which allowed us to create an Illumina-specific basis matrix with more immune cell types. However, it still lacked a few immune cell types (gamma-delta T cells, naïve B cells, memory B cells). There was low to moderate correlation between estimated cellular proportions and measured cellular proportions (second column in **Figure R2**). In contrast, immunoStates has consistently high correlation with measured cell proportions (third column in **Figure R2**).

Collectively, these results demonstrate that our proposed one-size-fits-most single matrix has higher accuracy than creating different basis matrix for each microarray platform.